# Pillar[5]arene-catalyzed anti-Markovnikov halogenations through cationic intermediates stabilization in confined spaces

Tianyue Xu[1,3], Shengtian Lai[1,3], Manjaly J. Ajitha [2] ✉, Kuo-Wei Huang [2] ✉ & Ying-Yeung Yeung [1] ✉

The confinement of reactants within catalytic cavities is important for achieving efficient and selective chemical transformations. Macrocyclic organic covalent hosts, which mimic enzymatic environments, offer well-defined, tunable cavities capable of substrate accommodation. These robust and synthetically accessible hosts can be engineered into catalysts by functionalizing their rims, where substrate selectivity emerges from size- or shape-complementary binding. This positioning brings reactive sites into proximity with catalytic functional groups at the rims of covalent hosts, accelerating reactions. Cationic intermediates are pivotal in many chemical transformations, and stabilizing these reactive species within confined microenvironments could unlock unconventional selectivity for synthesizing high-value compounds. Despite this potential, leveraging the cavities of covalent hosts to stabilize and confine cationic intermediates for regioselective reactions remains underexplored. Here we report that the π-basic cavity of pillar[n]arenes can effectively stabilize bromiranium intermediates generated during olefin halogenation, confining them in a controlled microenvironment. This strategy overrides the intrinsic Markovnikov preference, enabling highly selective anti-Markovnikov halogenation. Furthermore, we extend this catalytic system to achieve size-selective anti-Markovnikov halogenation of olefins. This approach opens new pathways for selective transformations through the confinement of a cationic intermediate.

The confinement of reactants within a catalyst pocket is one of the key factors for achieving efficient and selective catalytic transformations[1,2]. Synthetic host molecules (sometimes regarded as artificial enzymes), such as macrocyclic organic covalent hosts and self-assembled molecular cage nanoreactors, offer well-defined cavities that are suitable for substrate accommodation in catalysis due to their restricted dimensions[3–13]. They have garnered significant interest as platforms for catalysis by acting as a noncovalent protective group or an anchoring template to offer various selective reactions. Among these, macrocyclic organic covalent hosts, which can also be considered as

[1]Department of Chemistry and State Key Laboratory of Synthetic Chemistry, The Chinese University of Hong Kong, Shatin, NT, Hong Kong, China. [2]Center of Excellence for Renewable Energy and Storage Technologies, KAUST Catalysis Platform, and Division of Physical Sciences and Engineering, King Abdullah University of Science and Technology, Thuwal, Saudi Arabia. [3]These authors contributed equally: Tianyue Xu, Shengtian Lai. ✉e-mail: ajitha.john@kaust.edu.sa; kuowei.huang@kaust.edu.sa; yyyeung@cuhk.edu.hk

organocatalysts[14,15], such as cyclodextrins (**CD**)[16], crown ethers[17], cucurbiturils (**CB**)[18], calixarenes[19], and pillarenes (**PA**)[20] have gained continuous attention. These covalent hosts are not only easy to synthesize and highly robust but also feature tunable cavities with persistent gates, enabling substrate accommodation without the need for specialized reaction conditions.

For catalytic applications, covalent host systems are typically functionalized with catalytic sites at their rims. Substrate selectivity is determined by the complementary fit between the host cavity and the size or shape of the guest substrate molecule. Reaction acceleration of the designated pathway can be achieved by positioning the substrate's reactive sites in close proximity to the rim-anchored catalytic groups (Fig. 1a). To ensure satisfactory reaction performance, a super-stoichiometric amount of covalent hosts is usually required to suppress product inhibition. Several reactions using this strategy have been documented in the literature[21–23]. A few studies have demonstrated that accommodating appropriately oriented and sized substrates within covalent host cavities can improve the reaction selectivity. A landmark work was carried out by Breslow and coworkers in which an excess amount of **CD** was used to encapsulate phenol so that the **CD** rim's oxygen could be brought into close proximity to the *para*-position of phenol, enhancing the selectivity of *para*-chlorination (Fig. 1b)[24,25]. Ogoshi and coworkers conducted a representative competition experiment where a **PA**-derived phase-transfer catalyst preferentially accommodated the less bulky linear olefin over the branched olefin, allowing the linear olefin to be positioned near the phosphonium permanganate at the rim of **PA** for a more efficient oxidation (Fig. 1c)[26]. Nau and coworkers documented the encapsulation of bicyclic azoalkanes by excessive **CB** and Ag(I) to enhance the selectivity in photodeazetization[27]. The **CB** rim's carbonyl was proposed to bring the Ag(I) near the azo group, improving the selectivity of 1,5-hexadiene (HD) over bicyclo[2.2.0]hexane (BCH) (Fig. 1d). Some relevant works by Rebek and co-workers used cavitands as stoichiometric reaction nano-vessels to accommodate aliphatic substrates in folded conformation, facilitating macrocyclization over oligomerization in water[28,29].

Cationic intermediates are involved in many reactions, and the stabilization of these intermediates is crucial for satisfactory reaction performance. Although very challenging, there is a strong aspiration to confine the cationic intermediates within the microenvironment in order to attain unconventional selectivity for the synthesis of valuable products that are nontrivial to obtain[1–3]. In a study by Scarso and coworkers, it was found that **PA** could accelerate the mono-allylation of primary amines with allyl bromide in $CDCl_3$. Based on NMR studies, the ratio of secondary/tertiary amine product was improved in the presence of **PA** (Fig. 1e). It was proposed that the cavity of **PA** might accelerate the $S_N2$ reaction by stabilizing the developing charge on the nitrogen atom through electrostatic interactions with its aromatic units[30].

Electrophilic reactions of unactivated olefins, such as halogenation, represent a fundamental class of transformations[31]. A classic example involves the formation of three-membered haliranium cationic intermediates (e.g., bromiranium **A** from a $Br^+$ source), followed by nucleophilic attack at the more substituted carbon, adhering to Markovnikov's rule as well-accepted textbook knowledge (Fig. 1f, bottom left)[32]. However, achieving anti-Markovnikov selectivity, which would unlock access to a distinct class of valuable compounds, has remained a challenge. Although several elegant strategies enable anti-Markovnikov hydro-functionalization through alternative mechanisms (e.g., radical or metal-catalyzed pathways)[33–39], direct intermolecular catalytic anti-Markovnikov electrophilic halogenation of unactivated alkenes remains an unsolved problem in synthetic chemistry. Overcoming this limitation could provide an alternative route to functionalized products that are non-trivial to access.

Here, we report a study of **PA**-catalyzed anti-Markovnikov bromination of unactivated olefins (Fig. 1f, bottom right). Inspired by the concept of stabilizing bromiranium by Lewis bases (e.g., species **A**)[40–42], we hypothesize that the π-basic cavity of **PA** would stabilize the cationic bromiranium intermediate in a confined microenvironment and selectively shield the more substituted carbon (typically favored in Markovnikov addition). Thus, the less substituted (and less active) carbon could be more accessible to the nucleophiles. This microenvironmental control could override the intrinsic Markovnikov preference, offering a strategy for accessing elusive anti-Markovnikov halogenated products. By exploiting the catalytic application of the confined π-basicity of **PA**, we aim to establish a paradigm shift in electrophilic functionalization, moving beyond typical rim-based covalent host catalysis toward cavity-controlled reaction engineering.

## Results

Our study was initiated using olefin **1a** and BzOH (**2a**) as the reaction partners and 1,3-dibromo-5,5-dimethylhydantoin (DBDMH) as the halogen source in methylcyclohexane (Fig. 2a). No reaction was observed in the absence of a catalyst. Several common base (e.g., triphenylphosphine sulfide) and acid (e.g., MsOH, copper salt) catalysts were investigated, yielding a product mixture with a preference for Markovnikov selectivity, consistent with typical literature findings. The use of dimethylpillar[5]arene **PA1** as a catalyst at room temperature afforded a moderate yield with a preference for the Markovnikov product **3aa'**. However, lowering the reaction temperature to –10 °C reversed the selectivity, favoring the anti-Markovnikov product with a 60:40 r.r. (**3aa**:**3aa'**) and a 30% yield. These outcomes imply that pillar[5]arene could catalyze the bromination but lacks sufficient steric hindrance to effectively block nucleophilic attack at the more substituted carbon of the bromiranium species. Pillar[5]arenes **PA2**–**PA4** having different substituents were examined, and moderate-to-good anti-Markovnikov selectivity was observed. To our delight, catalyst **PA5** bearing *n*-hexyl groups at the rim gave a good yield of **3aa** (82%) and excellent anti-Markovnikov selectivity (r.r. > 20:1). Solvent polarity influenced the reaction outcome. While a more polar solvent afforded a higher yield, it resulted in poorer r.r., which could be attributed to an enhanced background reaction occurring outside the host (see Supplementary Information, Fig. S5). Pillar[6]arene **PA6** with a larger cavity could not favor the formation of the anti-Markovnikov product **3aa**, indicating that the relatively large size pillar[6]arene might not be effective in encapsulating the bromiranium species. We also studied some common covalent hosts such as α-cyclodextrin **CD**, hexylcalix[4]arene **CA** and cucurbit[6]uril **CB**, but they were found to be ineffective in providing anti-Markovnikov selectivity (Fig. 2a).

A series of substrates was then investigated (Fig. 2b). The catalytic anti-Markovnikov bromination proceeded efficiently to give the desired products **3** in good-to-excellent yields and regioselectivity. Besides, various substituted benzoic acids were compatible with the catalytic protocol. In particular, olefins with trimethylsiloxyl group furnished products **3aa**–**3ae** in excellent regioselectivity (r.r. > 20:1). Other olefinic substrates with alkyl (**1b**), ether (**1c**–**1f**), ester (**1g**–**1i**), halogen (**1j** and **1k**), and ketone (**1l**) substituents were studied and good yield and regioselectivity of the corresponding anti-Markovnikov products **3b**–**3l** were generally obtained, showcasing the high functional group compatibility of this catalytic protocol. We have also studied substates **1m** and **1n** with bulky substituents and longer hydrocarbon chains, and they are compatible with the catalytic protocol to give **3m** (70%, r.r. > 20:1) and **3n** (76%, r.r. = 12:1).

The power of this catalytic system is further demonstrated in the anti-Markovnikov halogenation of olefinic amide **1o**. Typically, such substrates undergo facile intramolecular halocyclization[43–45] to form six-membered rings[46] because this pathway is highly favored entropically. In our control experiment, halocyclization of **1o** proceeded exclusively with triphenylphosphine as the Lewis base catalyst, giving

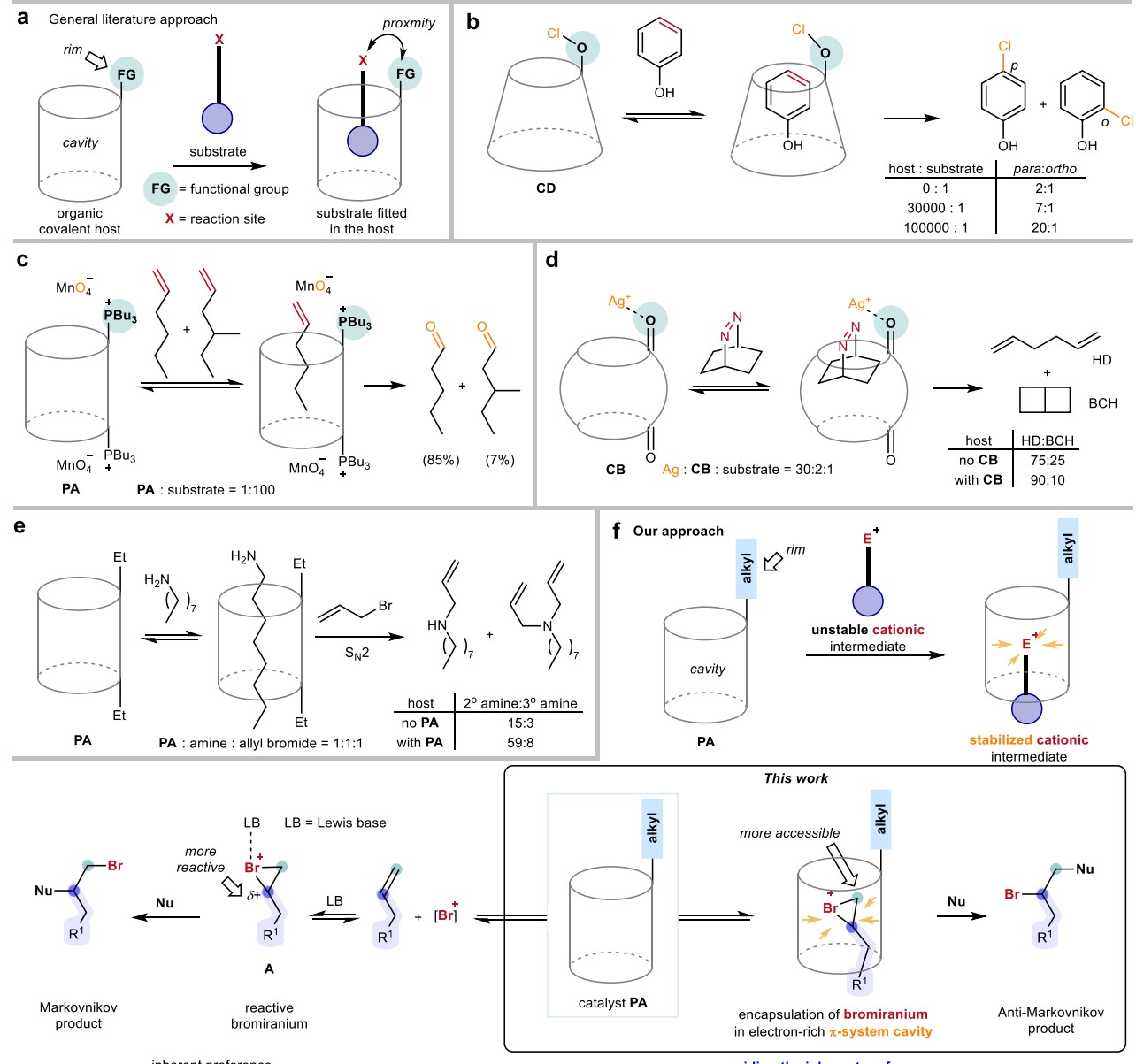

**Fig. 1 | Typical reactions promoted by macrocyclic organic covalent hosts and our approach. a** A common approach using rim's functional group of covalent hosts for catalysis. **b** Regioselective chlorination of phenol with cyclodextrin **CD**. **c** Size-selective oxidation with pillar[5]arene **PA**. **d** Photodeazetation of bicyclic azoalkanes with cucurbituril **CB**. **e** Mono-allylation of primary amine with pillar[5] arene **PA**. **f** Typical Markovnikov halogenation (bottom left) and our study on using pillar[5]arene **PA** to encapsulate bromiranium species for anti-Markovnikov halogenation (bottom right).

the six-membered ring Markovnikov selective lactam **4** (80%) even in the presence of BzOH as the external nucleophile (Fig. 2c). Outcompeting this inherent preference with an intermolecular reaction is exceptionally challenging. Nonetheless, with catalyst **PA5**, a 1:1 mixture of **1o** and BzOH undergoes completely selective intermolecular anti-Markovnikov halogenation to furnish **3o** (78%, r.r. = 11:1) with no detectable cyclized product **4**, attributed to the unfavorable cyclization inside the host. To the best of our knowledge, this represents the first example where a stoichiometric nucleophile outcompetes such a favorable intramolecular cyclization.

The resultant products can readily be converted to valuable commodity chemicals (Fig. 2d). For instance, the anti-Markovnikov products **3ba** and **3o** could easily be converted into allyl benzoates **5** and **6**, which are valuable building blocks[47,48]. Furthermore, aliphatic

azides **7**–**9**, valuable building blocks of detergents for membrane protein studies[49,50], were prepared from compound **3** in a single step using sodium azide. These methods establish straightforward and efficient alternatives to the typically tedious synthetic routes required for these important molecules.

Competition experiments were conducted to evaluate the site-selectivity. Substrates bearing two olefinic moieties on the same molecule were then examined in the intramolecular competition (Fig. 3a). Electrophilic halogenation of bulky olefins is favored by the inductive effect but disfavored by the steric effect. Thus, achieving regio and site-selective bromination of bis-olefinic substrates such as **10a** is challenging because multiple isomers could be formed. Indeed, when using triphenylphosphine sulfide as the catalyst, only 23% of **11a** was detected, and a mixture of bromination side products (~30%)

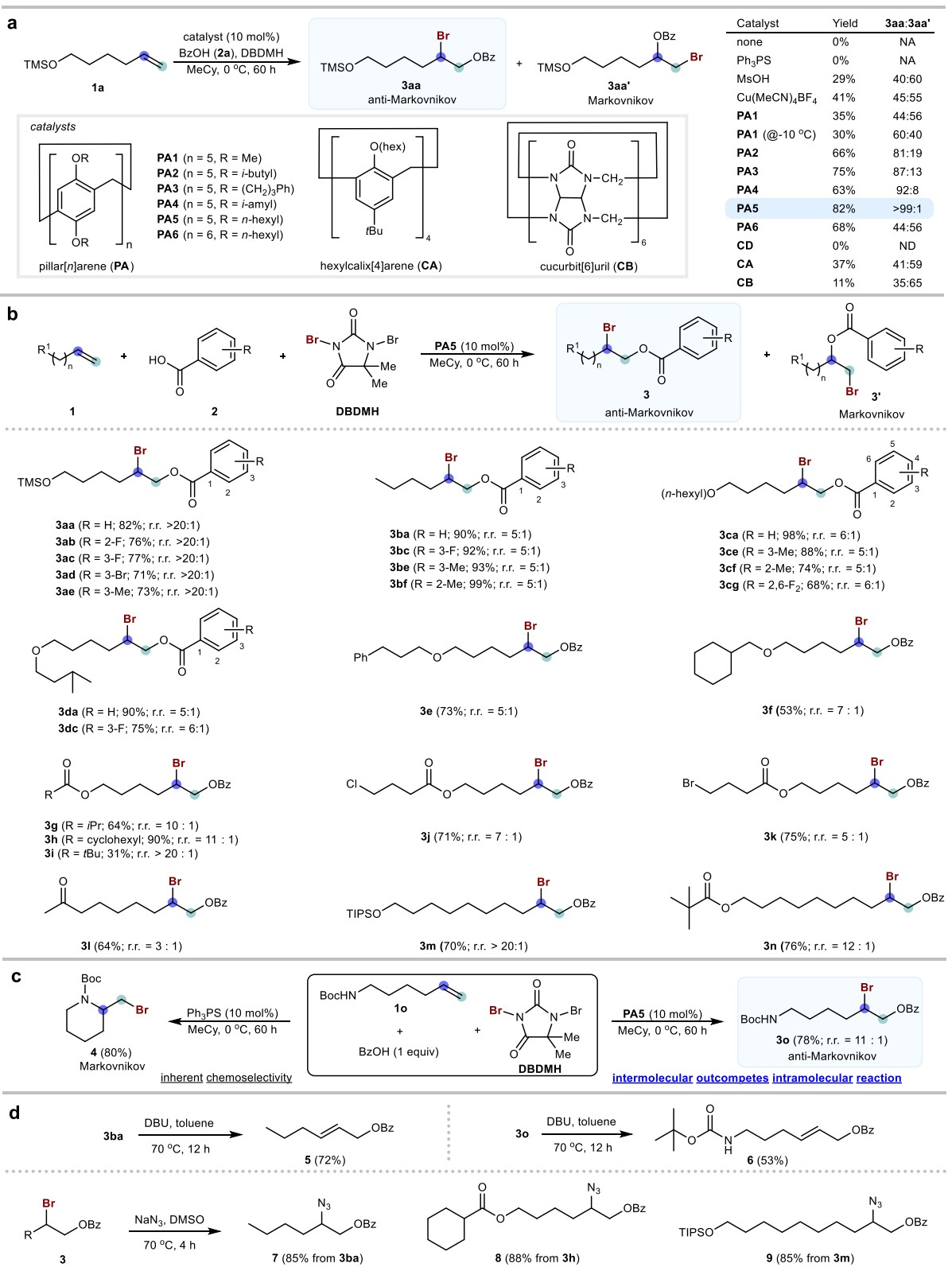

**Fig. 2 | Study on the anti-Markovnikov halogenation. a** Catalyst study. **b** Substrate scope. **c** Anti-Markovnikov halogenation of olefinic amide. **d** Synthetic utilities. Conditions: reactions were carried out with substrate **1** (0.1 mmol), carboxylic acid **2** (0.1 mmol), DBDMH (0.1 mmol), and catalyst (0.01 mmol, 10 mol %) in methylcyclohexane (0.35 mL) in the absence of light at 0 °C for 60 h. r.r. regioisomers ratio of **3:3'**. The yields are isolated yields.

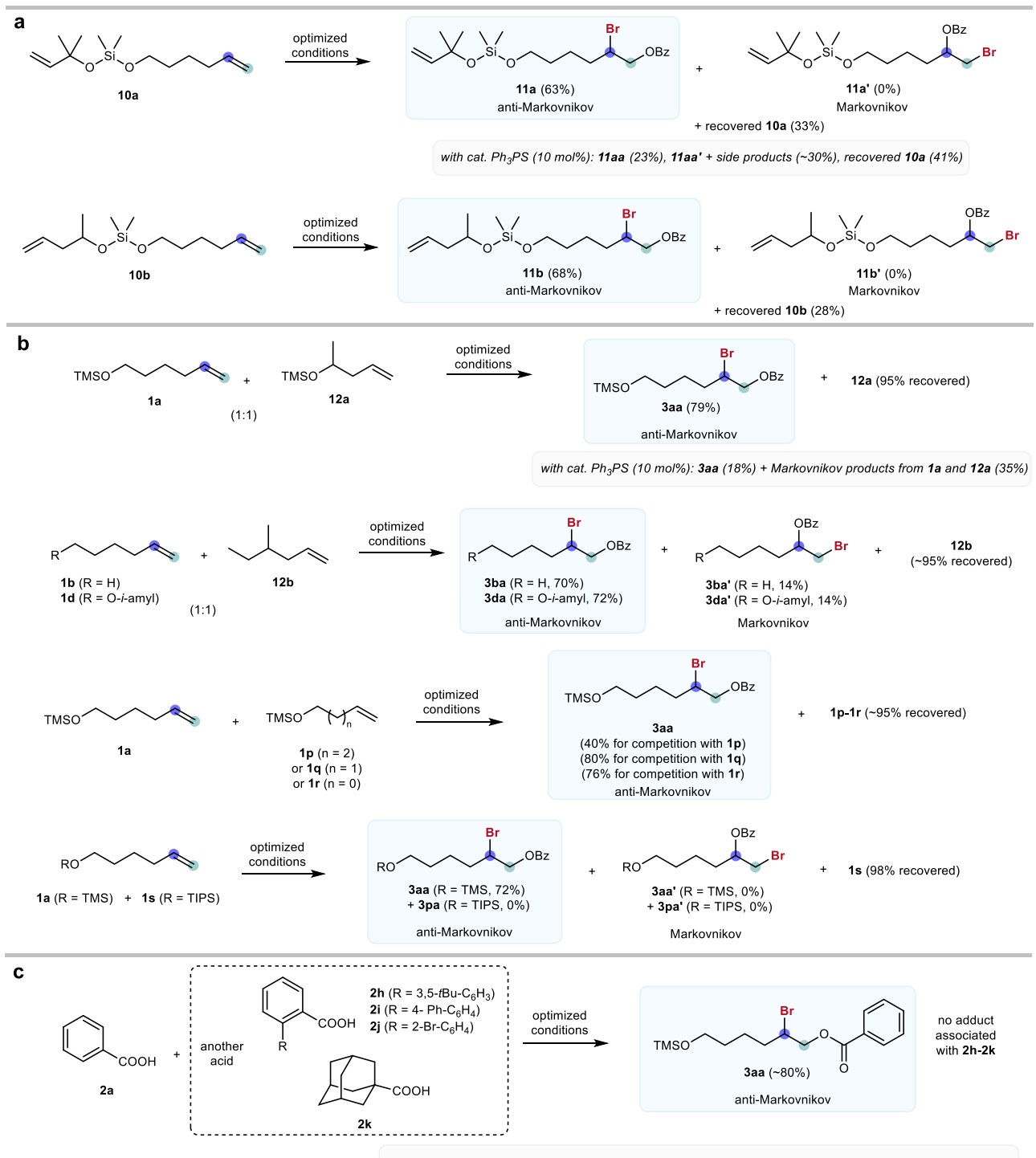

**Fig. 3 | Anti-Markovnikov halogenation in competition experiments.**
**a** Intramolecular competition experiments with substrates bearing two olefinic moieties on the same molecule. **b** Intermolecular competition experiments using different olefins. **c** Intermolecular competition experiments using different carboxylic acids. Optimized conditions: substrate (0.1 mmol), BzOH (0.1 mmol), DBDMH (0.1 mmol), and **PA5** (0.01 mmol, 10 mol%) in methylcyclohexane (0.35 mL) in the absence of light at 0 °C for 60 h. The yields are isolated yields.

together with 41% of unreacted substrate **10a** was observed (Supplementary Information, Fig. S1). In sharp contrast, site-selective anti-Markovnikov bromination product **11a** (63%) was obtained as the sole product when using catalyst **PA5**. Similarly, **11b** (68%) was obtained smoothly when using **10b** as the substrate. These results could be explained by the fact that the linear portion of the molecule could fit

into the cavity of **PA5** more readily to perform the anti-Markovnikov halogenation.

It was also found that the catalytic protocol was able to distinguish linear and branched olefinic substrates in the anti-Markovnikov halogenation (Fig. 3b). For instance, a 1:1 mixture of **1a** and **12a** was subjected to the study, and four products

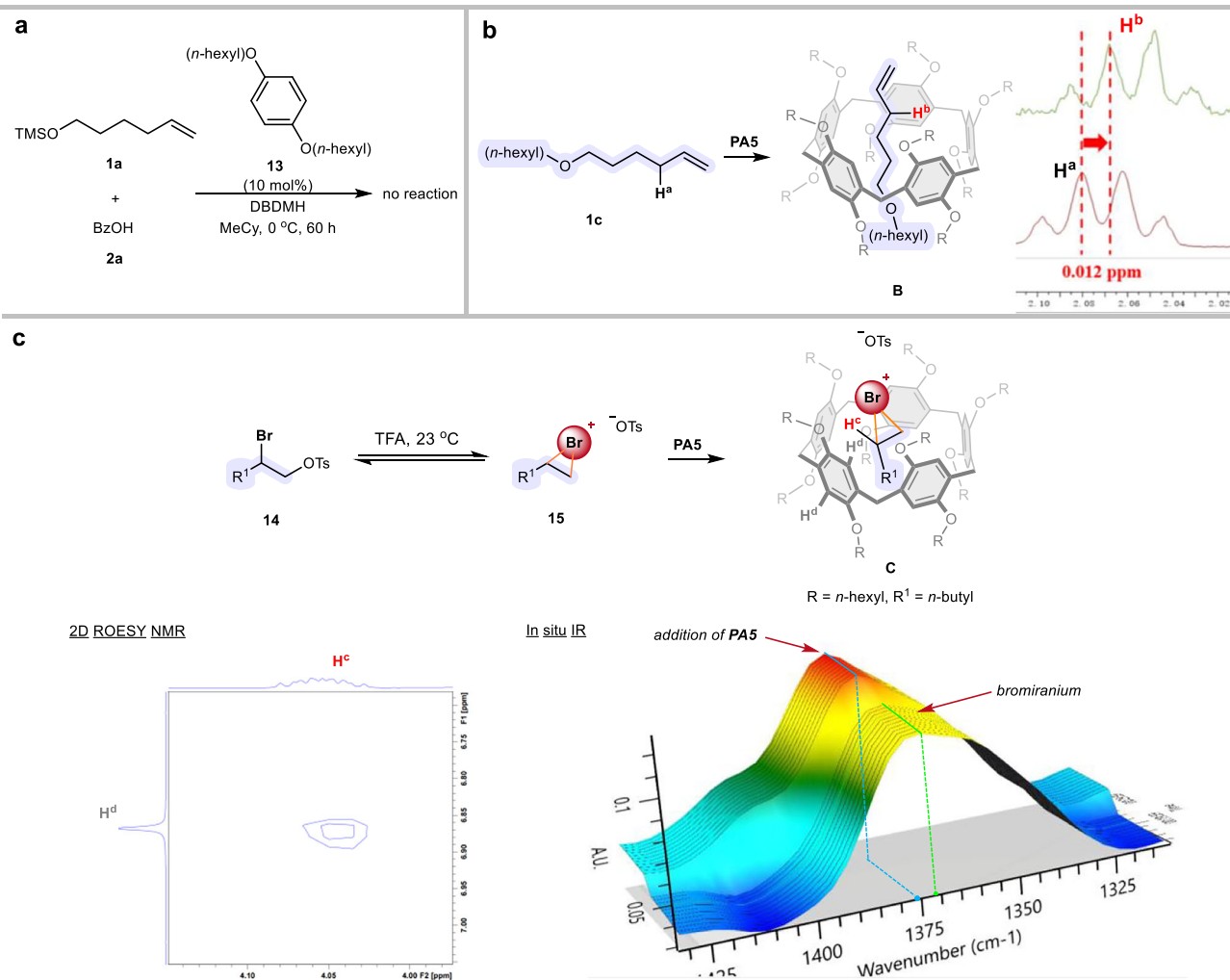

**Fig. 4 | Mechanistic studies. a** Control experiment using compound **13** as the catalyst. **b** NMR study on a mixture of **PA5** and **1c**. **c** 2D ROSEY NMR and in situ IR experiments to probe the encapsulated bromiranium species.

(anti-Markovnikov and Markovnikov products from both **1a** and **12a**) could be obtained in principle. However, **3aa** that was synthesized from the anti-Markovnikov halogenation of the linear olefin **1a** was obtained as the major product in good yield (79%). In contrast, when using the typical catalysts such as triphenylphosphine sulfide, a product mixture favoring Markovnikov products was obtained (Supplementary Information, Fig. S2). Competition experiments were also performed using other linear (**1b** or **1d**) and branched (**12b**) olefins, and products **3ba** and **3da** with good anti-Markovnikov selectivity were furnished.

Interestingly, the length of olefins also influenced the reactivity in this protocol. For the competition experiments on a 1:1 mixture of linear olefins **1a** (longer) and **1p–1r** (shorter), the longer substrate **1a** preferentially reacted to give the anti-Markovnikov product **3aa**, and bromination of the shorter substrates was not observed. Substrates bearing different sizes of silyl groups remote from the olefins could also be differentiated using this catalytic protocol. For example, when using triisopropylsilyl (TIPS) substrate **1s** in the competition experiment against **1a** bearing a relatively smaller TMS group, **3aa** was obtained as the sole product, and no bromination originating from **1s** was detected, attributed to the bulky TIPS group that disfavors the substrate accommodation (Supplementary Information, Figs. S15 and S16). However, substrate **1m** bearing TIPS but with a longer hydrocarbon chain could yield **3m** smoothly (Fig. 2b).

The catalytic protocol was also able to differentiate the size of nucleophilic partners. For example, a 1:1 mixture of benzoic acids **2a** and **2h** was subjected to the reaction with olefin **1a**, and benzoylated product **3aa** was obtained in 80% yield (Fig. 3c). In addition, no adduct corresponding to the bulkier benzoic acid was detected. In theory, the bulkier carboxylic acid, which is also a more electron-rich nucleophile, could be more reactive in the halogenation reaction. Indeed, in the benchmarking reaction using triphenylphosphine sulfide as the catalyst, a significant amount of products associated with **2h** (Marknovnikov and anti-Markovnikov) were detected (Supplementary Information, Fig. S3). Competition experiments using other bulkier carboxylic acids **2i–2k** against benzoic acid (**2a**) were studied and excellent size-selectivity of the nucleophilic partner was observed, giving **3aa** as the sole product.

A control experiment was carried out to shed light on the mechanism. Arene **13**, which is a key component of **PA5**, was used as a catalyst for the bromination, but no reaction was observed, suggesting that the pillar[5]arene cavity is necessary for the catalytic effect (Fig. 4a). NMR experiments were conducted to provide further mechanistic insight. When mixing **PA5** and the olefinic substrate **1c**, Hᵃ in **1c** shifted upfield, attributed to the encapsulation of **1c** in the shielded environment (Fig. 4b, species **B**). Job's plot was performed on the mixture **PA5** and **1c,** and the results indicated that the binding stoichiometry between **PA5** and **1c** is 1:1

(Supplementary Information, Fig. S8). However, the binding constant (6.12 ± 0.24 M$^{-1}$) between substrate **1** and **PA5** was found to be small (Supplementary Information, Fig. S9), suggesting that substrate binding in **PA5** might not be the critical factor to promote the bromination.

Further studies were performed to probe the encapsulated bromiranium species, which is believed to be the key intermediate of the reaction. In order to generate a sufficient amount of unstable bromiranium species, we used bromotosylate **14** in acidic media to generate bromiranium species **15** in situ[51]. Thus, a 2D ROESY NMR experiment on a mixture of **14**/TFA/**PA5** was carried out, and H$^c$ (in **15**) and H$^d$ (in **PA5**) have considerable $^1$H–$^1$H long-range interaction (Supplementary Information, Fig. S11). An in situ IR experiment was also conducted to gain more information (Fig. 4c). Upon mixing **14** and TFA, a new signal at 1370 cm$^{-1}$ that could correspond to the bromiranium species **15** emerged. The addition of **PA5** to the bromiranium species led to a blue shift of the absorption (Supplementary Information, Fig. S12). While multiple factors could contribute to the blue shift, we believe that one of the key factors could be the bond contraction caused by the nuclear repulsions of the bromiranium species within the confined cavity of **PA5**.

Density functional theory (DFT) calculations were also performed to gain insights into the plausible mechanism. To address dispersion interactions between the reactants, all geometry optimizations were carried out with the long-range-corrected ωB97XD functional[52–54], and the 6-31G(d,p) basis set.[55] Single-point solvation energy correction (in methylcyclohexane) was conducted using Truhlar's SMD model.[56] Energies were further refined at the ωB97XD/6-311 + G(d,p) level of theory. The relative Gibbs energies ($\Delta G$) in kcal/mol are provided with respect to the infinitely separated reactants. Extensive conformational searches for the most stable geometry of pillar[5]arene were first conducted to locate the starting catalyst structure.

The energy profile of the reaction using pillar[5]arene **PA1** was first calculated (Fig. 5a and Supplementary Information, Figs. S20–S26). It was found that reaction components included **1a**, DBDMH, BzOH, and **PA1** assembled to give **PTS$^{PA1}$** with a free energy increased by 5.9 kcal/mol. The free energy profile indicates that the transfer of the bromonium ion (Br$^+$) from DBDMH to the encapsulated olefin (**1a**) proceeds via transition state **TS1$^{PA1}$** with an activation barrier ($\Delta G^{\ddagger}$) of 9.3 kcal/mol. This process leads to the formation of a transient bromiranium intermediate **IM1$^{PA1}$**, followed by a structural reorganization to give intermediate **IM2$^{PA1}$** with energy reduced by 6.5 kcal/mol (with respect to **IM1$^{PA1}$**). Finally, **IM2$^{PA1}$** undergoes nucleophilic ring-opening of the bromiranium species by the benzoate to give the highly stable anti-Markovnikov addition product **P$^{PA1}$** via **TS2$^{PA1}$** ($\Delta G^{\ddagger}$ = 8.4 kcal/mol). The pathway to the Markovnikov product via **TS3$^{PA1}$** was also evaluated, and the energy was found to be 1.0 kcal/mol higher than that of the anti-Markovnikov pathway via **TS2$^{PA1}$**. This could be attributed to the increased steric repulsion when the benzoate approaches the more shielded (and the more substituted) carbon for the inherent Markovnikov selectivity.

In sharp contrast, in the absence of pillarene, the formation of bromiranium species **IM1** ($\Delta G$ = 31.6 kcal/mol) by a bromonium transfer from DBDMH to the olefin **1a** in **PTS** ($\Delta G$ = 9.6 kcal/mol) was found to be highly unfavorable (Fig. 5b and Supplementary Information Fig. S27). Thus, the pillarene-catalyzed anti-Markovnikov halogenation is favorable as indicated by the smaller barrier compared to the non-encapsulated bromiranium species and the exergonic nature.

Next, we analyzed the bromiranium cationic species in the optimal catalyst **PA5**. Interaction of DBDMH and BzOH with olefin **1a** in the cavity of **PA5** is promoted to give **PTS$^{PA5}$** ($\Delta G_{rel}$ = 1.9 kcal/mol), followed by the formation of bromiranium species **IM1$^{PA5}$** ($\Delta G_{rel}$ = 6.0 kcal/mol) (Fig. 5b and Supplementary Information

Fig. S17). The close resemblance of the structure and energy of **PTS$^{PA5}$** and **IM1$^{PA5}$** (from **PA5**) and that of species **PTS$^{PA1}$** and **IM1$^{PA1}$** (from the less bulky **PA1**) provided us with strong confidence that the smaller system used to study the reaction mechanism is adequate. Similar to the situation in **IM1$^{PA1}$**, the less substituted carbon of the bromiranium moiety in **IM1$^{PA5}$** is more exposed to the rim of **PA5**, and therefore, more accessible by the benzoate nucleophile to give the experimentally observed anti-Markovnikov product. It was realized that the bromiranium species is considerably more stabilized in **PA5** than **PA1**, indicated by the more significant reduction in free energy between **IM1/IM1$^{PA5}$** ($\Delta G_{rel}$ = −25.6 kcal/mol) as compared with that between **IM1/IM1$^{PA1}$** ($\Delta G_{rel}$ = −18.5 kcal/mol). This finding is also consistent with the experimental results in which **PA5** exhibited a better catalytic efficiency than **PA1** (Fig. 2a).

IBSI[57], NBO[58], and AIM analysis[59,60] revealed that the bromiranium cationic species engages favorably inside the cavity of **PA5** via multiple non-covalent interactions (Fig. 5c and Supplementary Information Figs. S18 and S19). Specifically, a significant C = C(π)→Br−C(σ*) interaction between the phenyl group of **PA5** and the bromiranium cation was observed. In addition, a number of C−H hydrogen bond interactions [C = C(π) → H−C(σ*) and O(LP) → H−C(σ*)] between **PA5**'s phenyl ether groups and the bromiranium's C−Hs were found to be crucial in the stabilization. An energy decomposition analysis (sobEDA and sobEDAw)[61] was also performed to provide more insight into the interaction between **PA5** and the reactants (Supplementary Information, Table S1), and it suggests that while electrostatic interaction primarily drives the formation of the bromiranium ion in **PA5**, dispersion forces also make a substantial contribution to the stabilization of this host-guest complex.

Based on the abovementioned studies, pillarene could serve two key functions in the anti-Markovnikov halogenation: (1) catalyzing bromination by fitting the substrate with an appropriate size and stabilizing the bromiranium cationic intermediate within its π-basic cavity and (2) encapsulating the intermediate to enforce anti-Markovnikov selectivity through spatial confinement.

In summary, we have developed a supramolecular strategy utilizing the π-basic cavity of pillarenes to stabilize bromiranium intermediates, enabling selective shielding of the more substituted carbon and redirecting nucleophilic attack toward the less substituted position. This catalytic approach successfully achieves anti-Markovnikov bromination, demonstrating how microenvironmental control can override intrinsic reactivity biases to access challenging products. While covalent macrocyclic hosts have been used as nanoreactors to accelerate reactions, their use in stabilizing cationic intermediates for selective transformations remains in its infancy. We anticipate that this research will inspire further research into the design of tailored host systems for selective reaction control, opening new avenues in synthetic chemistry.

## Methods

### General procedure for anti-Markovnikov intermolecular bromoesterification

To a stirred solution of carboxylic acid **2** (0.1 mmol, 1 equiv), catalyst **PA5** (0.01 mmol, 10 mol%) in methylcyclohexane (0.35 mL) was added olefinic substrate **1** (0.1 mmol, 1 equiv). The solution was stirred at 0 °C and DBDMH (0.1 mmol, 1 equiv) was added. The resultant mixture was stirred for 60 h at the same temperature and quenched with a saturated aqueous Na$_2$SO$_3$ solution. The organic phase was separated, and the aqueous phase was extracted with CH$_2$Cl$_2$ (2 mL × 3). The combined organic phase was washed with brine (3 mL), dried over anhydrous Na$_2$SO$_4$, filtered, and concentrated under reduced pressure. The residue was purified over silica gel column chromatography (EtOAc:hexanes, 1:100–1:20) to afford the desired anti-Markovnikov halogenation product **3**.

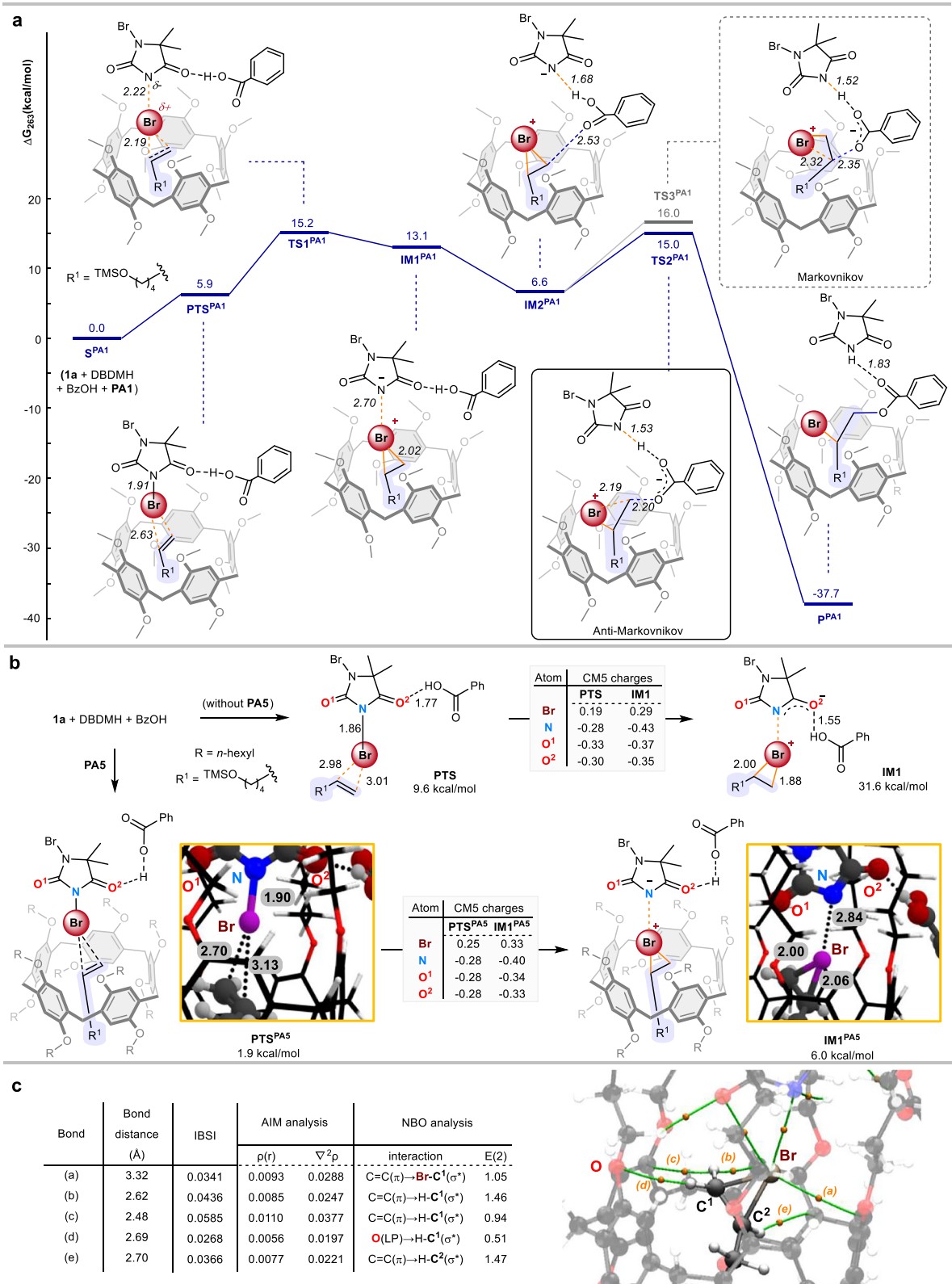

**Fig. 5 | Computational studies. a** Energetics of the formation of bromiranium species inside pillar[5]arene **PA1** and the anti-Markovnikov halogenation reaction. $\Delta G_{rel}$ values are given in kcal/mol. **b** Calculated structure of the bromiranium species inside and outside pillar[5]arene **PA5**. **c** IBSI, AIM, and NBO analyses on the encapsulated bromiranium species.

## Data availability

Experimental details, characterization of the compounds, spectral data, and theoretical-calculation results are available within the published manuscript and supplementary information. The X-ray crystallographic data for the structure reported in this study have been deposited at the Cambridge Crystallographic Data Center (CCDC) under deposition number CCDC 2504582 (for **28**). These data can be obtained free of charge from the Cambridge Crystallographic Data Center via www.ccdc.cam.ac.uk/data_request/cif. All data are available from the corresponding author upon request.

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

## Acknowledgements

This study was supported by the Distinguished Young Scholar Fund from the National Natural Science Foundation of China (grant no. NSF22425110), Hong Kong Special Administrative Region General Research Funding (grant no. CUHK14307122), the Chinese University of Hong Kong Direct Grant (grant no. 4053329), and Innovation and Technology Commission to the State Key Laboratory of Synthetic Chemistry. The authors also acknowledge financial support and the service of Ibex, Shaheen 2 High Performance Computing Facilities from King Abdullah University of Science and Technology.

## Author contributions

K.-W.H. and Y.-Y.Y. conceived of and directed the project. T.X. and S.L. contributed equally to this work, and they performed the investigation, synthesis, and data analysis. M.J.A. performed the DFT calculations and analyzed the calculation results. T.X., S.L., M.J.A., K.-W.H., and Y.-Y.Y. co-wrote the manuscript.

## Competing interests

The authors declare no competing interests.
