## [Transparent Peer Review file · Nature Communications]

Pillar[5]arene-Catalyzed Anti-Markovnikov Halogenations Through Cationic Intermediates Stabilization in Confined Spaces

Corresponding Author: Professor Ying-Yeung Yeung

Version 0:

Reviewer comments:

Reviewer #1

(Remarks to the Author)

This work presents a comprehensive study of the "Pillar[5]arene-Catalyzed Anti-Markovnikov Halogenations" reaction, and the mechanistic elucidation is thorough and well-supported. The authors have provided clear and well-substantiated responses to all three reviewers' comments, supplemented with appropriate additional experimental data.

In its current form, the manuscript is considered suitable for publication. Nevertheless, it is suggested that the following two minor revisions could further improve the clarity and presentation of the manuscript:

1. In line with Reviewer 3's Comment 7 regarding the limited familiarity of 1,3-dibromo-5,5-dimethylhydantoin (DBDMH) to a general audience, it is recommended that the chemical structure of DBDMH be included in Figure 2b where the reaction scheme is shown.
2. Also noted is the point raised in Reviewer 3's Comment 9 concerning the small size of the IR spectra in Figure 4. Enlarging the size of the spectra would facilitate the observation of spectral details and enhance the overall visual presentation.

Reviewer #2

(Remarks to the Author)

The authors report very interesting results on the regioselective opening of bromonium ions catalyzed by pillar[5]arene. High regioselectivities (>20:1) are obtained when the substrates contain a bulky terminal group (OSiR₃ or t-Bu) on one side, whereas in its absence the selectivity decreases to approximately 5:1. The control experiments convincingly demonstrate that the reaction occurs within the pillar[5]arene cavity.

The results appear to be well documented in the Supporting Information and are supported by DFT calculations. Overall, these findings are highly compelling and provide an elegant entry into a substance class that is otherwise difficult to access by alternative methods.

I recommend only minor revisions.

1) Proper citation of related pillararene work; for instance. <https://pubs.acs.org/doi/10.1021/acscatal.4c04836> ; a similar suggestion was made by former Ref. 2. Surprisingly, nothing was added. The following sentence would be appropriate: "Unlike conventional strategies that rely on rim-functionalized covalent hosts to catalyze reactions, ..."

2) Adding details to Figures on how the yield were determined. Clearly stating if it is NMR yield (How determined? Internal standard?) or isolated yield

Reviewer #3

(Remarks to the Author)

The manuscript by Xu et al. is a resubmission of a previous version, reviewed for [redacted], and has undergone

major revision. It reports a regioselective anti-Markovnikov bromination of alkenes catalyzed by substituted pillararene hosts. The authors have carefully addressed the reviewers' comments and performed substantial additional work, which, in my opinion, has improved the quality and clarity of the manuscript. Overall, I am satisfied with the revised version. However, I do not fully agree with the authors' claim that the anti-Markovnikov selectivity (regioselectivity) is determined by factors other than substrate fitting within the host. The experimental data clearly demonstrate that efficient catalysis and selectivity require appropriate substrate (intermediate) positioning. Shorter substrates show poor selectivity, while bulkier substrates do not react, and macrocycles bearing shorter rim substituents display reduced catalytic efficiency and selectivity. Certainly, an important advantage of the authors' system is the high catalytic performance achieved with sub-stoichiometric amounts of the host, indicating the absence of significant product inhibition. This contrasts with many previous reports where stoichiometric host loadings are used to suppress product inhibition. In this context, the statement that excess host is required "to shift the equilibrium in favor of the guest/host complex" is misleading, as such systems typically exhibit very high host-guest binding constants. After addressing these minor clarifications, I believe the manuscript will be suitable for publication.

Reviewer #4

(Remarks to the Author)

In this manuscript, Huang, Yeung, and co-workers report on the unprecedented regioselectivity achieved in the anti-Markovnikov oxy-halogenation of olefins using a pillararene host.

I have reviewed a previous iteration of this work prior to its current submission. I am pleased to observe that the authors have comprehensively and satisfactorily addressed the constructive comments raised during the previous review process, including my own. The manuscript is now significantly clearer and presents a focused narrative regarding the reactivity of alkene in the unprecedented regioselective halogenation catalyzed by pillar[5]arenes. The data analysis, methodological approach, and the interpretation of the experimental results are rigorous and solid.

The key findings regarding the steering of regioselectivity are of high significance to the scientific community. This work effectively highlights the potential of macrocyclic hosts in controlling reaction outcomes and will undoubtedly stimulate further investigations in the field.

Overall, I consider the manuscript suitable for publication. I have only one minor revision regarding references and the correct contextualization of the study.

Specific Comments: I ask the authors to rephrase the sentence: "Here we report a proof-of-concept study demonstrating PA-catalyzed...". It is important to give proper credit to relevant prior art in this domain. Specifically, the authors should cite ACS Catal. 2024, 14, 15850–15857. This publication is highly relevant as it demonstrated the first application of a pillar[5]arene as a supramolecular catalyst for SN2 reactions, achieved by stabilizing the reaction transition state within the cavity. Acknowledging this work will provide a more accurate historical context for the current study.

Reviewer #6

(Remarks to the Author)

[Editor's note: Reviewer 5 could not continue the review process with us. As such, Reviewer 6 was recruited to assess the authors' response to Reviewer 5's concerns.]

According to the editor's requirement, I have focused my evaluation on the computational aspects of the revised manuscript. The theoretical work is performed and presented to a very high standard and significantly enhances our understanding of the role of pillar[5]arene in controlling the reaction selectivity. The revised version has fully addressed the issues raised by previous reviewers. Herein, I would like to recommend its publication.

Thank you very much for spending time reviewing the manuscript "**Pillar[5]arene-Catalyzed Anti-Markovnikov Halogenations Through Cationic Intermediates Stabilization in Confined Spaces**". We thank the reviewers for their constructive comments and suggestions, which are important in improving the manuscript. The manuscript has been revised accordingly (the changes are yellow-highlighted in separate files in the attachments), and the corrections are listed below:

Reviewer 1

Comment 1

“In line with Reviewer 3’s Comment 7 regarding the limited familiarity of 1,3-dibromo-5,5-dimethylhydantoin (DBDMH) to a general audience, it is recommended that the chemical structure of DBDMH be included in Figure 2b where the reaction scheme is shown.”

Response:

We thank the comment by the reviewer. We agree that the structure of DBDMH should be shown in Figure 2. We have now included the structure of DBDMH in Figure 2b.

Comment 2

“Also noted is the point raised in Reviewer 3’s Comment 9 concerning the small size of the IR spectra in Figure 4. Enlarging the size of the spectra would facilitate the observation of spectral details and enhance the overall visual presentation.”

Response:

We thank the comment by the reviewer. We have now enlarged the size of the IR spectra to show better visual presentation in Figure 4.

Reviewer 2

Comment 1

“Proper citation of related pillararene work; for instance. <https://pubs.acs.org/doi/10.1021/acscatal.4c04836>; a similar suggestion was made by former Ref. 2. Surprisingly, nothing was added. The following sentence would be appropriate: "Unlike conventional strategies that rely on rim-functionalized covalent hosts to catalyze reactions, ...”

Response:

We thank the comment by the reviewer. We have now added the reference and additional description on page 3 about the study by Scarso and coworkers.

Comment 2

“Adding details to Figures on how the yield were determined. Clearly stating if it is NMR yield (How determined? Internal standard?) or isolated yield”

Response:

The yields are isolated yields. We have now added this information to the figure captions of Figure 2 and Figure 3.

Reviewer 3

Comment 1

“I do not fully agree with the authors’ claim that the anti-Markovnikov selectivity (regioselectivity) is determined by factors other than substrate fitting within the host. The experimental data clearly demonstrate that efficient catalysis and selectivity require appropriate substrate (intermediate) positioning. Shorter substrates show poor selectivity, while bulkier substrates do not react, and macrocycles bearing shorter rim substituents display reduced catalytic efficiency and selectivity ...”

Response:

We thank the reviewer’s comment. We agree that efficient catalysis and selectivity require appropriate positioning of the cationic intermediate within the host. A sentence “catalyzing bromination by fitting the substrate with an appropriate size and stabilizing the bromiranium cationic intermediate within its π -basic cavity” has now been added on page 11.

Comment 2

“Certainly, an important advantage of the authors’ system is the high catalytic performance achieved with sub-stoichiometric amounts of the host, indicating the absence of significant product inhibition. This contrasts with many previous reports where stoichiometric host loadings are used to suppress product inhibition. In this context, the statement that excess host is required “to shift the equilibrium in favor of the guest/host complex” is misleading, as such systems typically exhibit very high host–guest binding constants”

Response:

We thank the reviewer for the comment. We have modified the sentence on page 3 as “a super-stoichiometric amount of covalent hosts is usually required to suppress product inhibition”.

Reviewer 4

Comment 1

“I ask the authors to rephrase the sentence: “Here we report a proof-of-concept study demonstrating PA-catalyzed...”. It is important to give proper credit to relevant prior art in this domain. Specifically, the authors should cite ACS Catal. 2024, 14, 15850–15857. This publication is highly relevant as it demonstrated the first application of a pillar[5]arene as a supramolecular catalyst for SN2 reactions, achieved by stabilizing the reaction transition state within the cavity. Acknowledging this work will provide a more accurate historical context for the current study...”

Response:

We thank the comment by the reviewer. We have now added the reference and additional description on page 3 about the study by Scarso and coworkers.

Reviewer 6

Comment 1

“According to the editor's requirement, I have focused my evaluation on the computational aspects of the revised manuscript. The theoretical work is performed and presented to a very high standard and significantly enhances our understanding of the role of pillar[5]arene in controlling the reaction selectivity. The revised version has fully addressed the issues raised by previous reviewers. Herein, I would like to recommend its publication...”

Response:

We appreciate the comments by the reviewer. We have further revised the manuscript according to the comments suggested by other reviewers. We believe that this revised version of the manuscript should be suitable for publication.

Thank you for your attention.

Yours sincerely,

Ying-Yeung YEUNG
Department of Chemistry
The Chinese University of Hong Kong
Shatin, NT
Hong Kong
Phone: (852) 39436377
yyyeung@cuhk.edu.hk